# Task-Circuit Quantization: Leveraging Knowledge Localization and Interpretability for Compression

**Hanqi Xiao, Yi-Lin Sung, Elias Stengel-Eskin, Mohit Bansal**
UNC Chapel Hill

## Abstract

Post-training quantization (PTQ) reduces a model's memory footprint by mapping full precision weights into low bit weights without costly retraining, but can degrade its downstream performance especially in low 2- to 3-bit settings. We develop a new mixed-precision PTQ approach, Task-Circuit Quantization (TACQ), that draws parallels to automated circuit discovery, directly conditioning the quantization process on specific weight circuits – which we define as sets of weights associated with downstream task performance. These weights are kept as 16-bit weights, while others are quantized, maintaining performance while only adding a marginal memory cost. Specifically, TACQ contrasts unquantized model weights with a uniformly-quantized model to estimate the expected change in weights due to quantization and uses gradient information to predict the resulting impact on task performance, allowing us to preserve task-specific weights. We compare TACQ-based quantization to existing mixed-precision quantization methods when conditioning both on general-purpose and task-specific data. Across QA, math reasoning, and text-to-SQL tasks for both Llama-3 and Qwen2.5, we find that TACQ outperforms baselines using the same calibration data and a lower weight budget, achieving major improvements in the 2 and 3-bit regime. With only 3.1 bits we are able to recover 96% of Llama-3-8B-Instruct's unquantized 16-bit MMLU performance, obtaining a 5.25% absolute improvement over SPQR. We also observe consistently large gains over existing methods in the 2-bit regime, with an average gain of 14.74% over the strongest baseline, SliM-LLM. Moreover, we observe a 7.20% gain without conditioning on specific tasks, showing TACQ's ability to identify important weights is not limited to task-conditioned settings.[1]

## 1 Introduction

Despite the broad range of applications for large language models (LLMs) (Yang et al., 2024; Sallam, 2023), their adoption is often hindered by their computational cost and memory footprint. This is an especially important consideration in domains where models need to be run locally (i.e. given privacy constraints, for example, in processing patient records), or where users are latency or compute-constrained, e.g. providing real-time customer service or when running on edge hardware (Jiayi et al., 2023; Rome et al., 2024; Friha et al., 2024). Furthermore, in many cases, models are tailored to specific use cases; for example, a model might be specialized for tasks like question-answering or translating text to SQL.

Post-training quantization (PTQ) has emerged as a promising way to reduce the memory footprint of large models, efficiently compressing large pre-trained models without costly retraining by storing model weights at lower precision, reducing memory consumption by 2-4x (Frantar et al., 2023a). Many attempts have been made to increase the amount of compression that is possible, but current processes are bottle-necked at 4-bit compression, with notable performance degradation at 2- and 3-bit precision (Huang et al., 2024b; Kim et al., 2024). Nearly all PTQ methods rely on a small mini-batch of general-purpose pre-training data samples that accounts for changes in activations due to quantization. Similarly, past work

---

[1]Code: https://github.com/The-Inscrutable-X/TACQ.

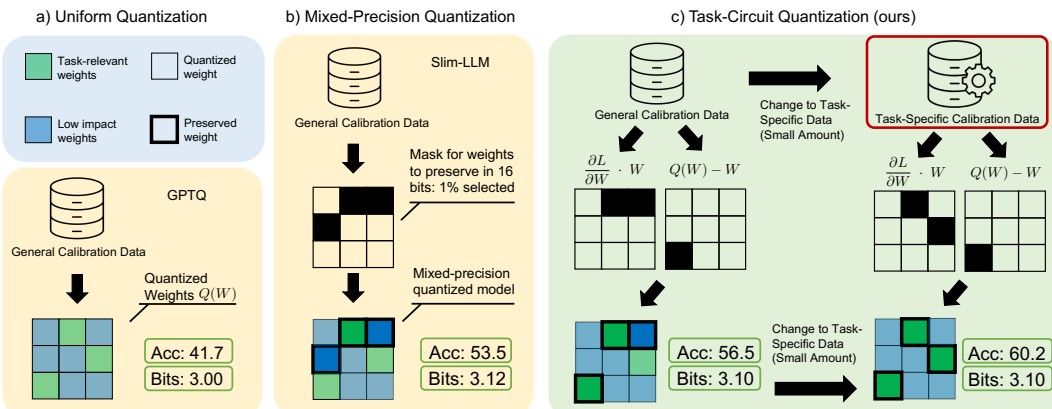

Figure 1: By leveraging task conditioning and task-circuit quantization, we are able to drastically improve performance when conditioning on both general-purpose and task-specific data. Accuracy (%) refers to one task (the "Humanities" split in the MMLU (Hendrycks et al., 2021) dataset), evaluated at a 3-bit compression setting.

has proposed using a general calibration set for mixed-precision quantization, preserving important "outlier" weights in a higher bit-width (typically 16 bits) while compressing the remaining weights, achieving better performance while minimally increasing storage cost (Dettmers et al., 2023; 2022).

Indeed, not all weights in a given LLM are equally important for a given task, and information density in LLMs is low, as indicated by the fact that parts of models can be removed outright and small subnetworks can be trained to recover the original LLM's performance (Ma et al., 2023; Men et al., 2024). Furthermore, research attempting to provide a mechanistic understanding of LLMs by studying activation circuits has revealed evidence of task-specific weights in LLMs, where small subsets of weights in an LLM are especially important for specific tasks (Christ et al., 2024; Olah et al., 2020; Kramár et al., 2024; Syed et al., 2023; Dai et al., 2022). We first adapt existing mixed-precision quantization methods to task-specific settings, finding that conditioning on task data generally helps performance but that quantized models continue to fall short of their unquantized versions, especially in 2-bit settings. To close this gap, we draw parallels to methods in automated circuit discovery (Syed et al., 2023; Conmy et al., 2023), input attribution (Shrikumar et al., 2017; Binder et al., 2016; Selvaraju et al., 2020), and localization in model editing (Meng et al., 2023) to create an improved saliency metric that accounts for both global gradient information and the effects of quantization in particular. We employ this saliency metric to localize a small number of highly important weights, preserving these weights in uncompressed 16-bit form during quantization.

Specifically, our saliency metric estimates the impact of quantization on weights by contrasting the original model "clean model" with a naively-quantized model "corrupt model". We combine this impact metric with global gradient-based saliency information to identify weights that are the most critical to task performance. Our approach can intuitively be thought of as efficiently estimating the impact on the loss if we were to remove a weight i.e., a weight's general importance to a circuit, scaled by how much we expect quantization to change its raw value, revealing which weights are most crucial to a task, as seen in Fig. 1.

We evaluate TACQ's ability to compress models to 2- to 3-bit size on multiple-choice question-answering benchmarks such as MMLU (Hendrycks et al., 2021), a math benchmark, GSM8k (Cobbe et al., 2021), as well as Spider (Yu et al., 2019), a standard text-to-SQL generation benchmark. We see improvements across all settings compared to state-of-the-art mixed-precision PTQ baselines like SqueezeLLM (Kim et al., 2024), SPQR (Dettmers et al., 2023), and SliM-LLM (Huang et al., 2024b) using the same conditioning data and at matched memory budgets. TACQ especially excels in 2-bit settings, where we improve accuracy over the strongest baseline, SliM-LLM (Huang et al., 2024b), by 15.99% (absolute)

on GSM8k, 21.92% on Spider, and 14.35% on MMLU. Moreover, TACQ also provides consistent improvements in 3-bit settings, and we show that its improvements are consistent across different quantization budgets and model sizes. Scaling to 32B models, TACQ recovers 87.93% of the 16-bit unquantized model's MMLU performance at 2-bit quantization. Furthermore, while TACQ obtains the largest gains over baselines in the task-specific setting (i.e. where all methods are conditioned on task-specific data), it also outperforms baselines when using general-purpose pretraining data for calibration, obtaining a 7.20% gain on the MMLU Humanities Split over the strongest baseline SliM-LLM in 2-bits. We also find that previous mixed-precision methods require searching for thresholds to specify bit-width (Dettmers et al., 2023; Kim et al., 2024) or face difficulty adjusting bit-width in a fine grained manner (Huang et al., 2024b). Our method only has one hyperparameter, the fraction of outliers to preserve in 16-bits, which allows us to specify the desired compression ratio deterministically.

## 2 Background and related work

**Quantization.** LLM weights are stored as 2D tensors of 16-bit floats, the fundamental uniform quantization approach (channel-wise linear quantization) compresses these weights by considering each row (or channel) independently, capturing the max and min of the weight values to adaptively map floats to integers. Specifically, for a row of weights $w \in W$, a target compression bit-width $N$, the quantized weight $Q(w)$ is defined as:

$$\Delta = \frac{w_{\max} - w_{\min}}{2^N - 1}, \ z_0 = \text{round}\left(-\frac{w_{\min}}{\Delta}\right), \ Q(w) = \text{clip}\left(\text{round}\left(\frac{w}{\Delta} + z_0\right), 0, 2^N - 1\right) \quad (1)$$

The possible values of weights after quantization is referred to as gridlines.

**GPTQ based Quantization.** A standard framing of the quantization problem is layerwise reconstruction, seeking to minimize layer-wise reconstruction loss $L = ||WX - Q(W)X||_2^2$ after quantization (Frantar et al., 2023a; Huang et al., 2024b; Dettmers et al., 2023), where $X$ is the layer input. Our method builds on GPTQ quantization (Frantar et al., 2023a), which quantizes weight matrices one column at a time and makes per row adjustments to the values of unquantized weights to compensate for error induced by quantization. Specifically, given the weight to quantize $w_q$ at row index $q$ and the approximate Hessian $H = XX^T$ of the loss with respect to the inputs, we define the adjustments $dw$ to unquantized weights as:

$$dw = -\frac{w_q - Q(w_q)}{\left[H_F^{-1}\right]_{qq}} \cdot \left(H_F^{-1}\right)_{:,q} \quad (2)$$

where $F$ denotes the set of unquantized weights. We focus on this type of training-free post-training quantization for its efficiency and widespread adoption, further providing a complete derivation and descriptions of other quantization techniques in Appendix D.

**Mixed-Precision Quantization.** Many existing quantization methods (Frantar et al., 2023a; Shao et al., 2024a; Lin et al., 2024; Ashkboos et al., 2024) quantize models to a fixed precision for all weights. Mixed-precision methods (Kim et al., 2024; Dettmers et al., 2023; Huang et al., 2024b; Dettmers et al., 2022) assign bit-width based on the relative importance of weights to achieve higher performance. The line of work we follow preserves highly sensitive weights, referred to as *outliers*, at higher precision (Dettmers et al., 2023; Kim et al., 2024). Prior work has demonstrated that keeping just 1% of these outliers unquantized can substantially improve model accuracy (Dettmers et al., 2023). Let $\theta$ be the parameter set of the model and $\theta_{\text{outliers}} \subset \theta$ the parameters we keep unquantized. The parameter set of the mixed-precision quantized model $\theta_q$ can be formally defined as:

$$\theta_q = \{Q(w); \ \forall w \in \theta_{\text{normal}}\} \cup \theta_{\text{outliers}} \quad (3)$$

where $\theta_{\text{normal}} = \theta \setminus \theta_{\text{outliers}}$. Previous approaches have defined weight importance based on local layer-wise loss and weight magnitude (Dettmers et al., 2023; Huang et al., 2024b), other

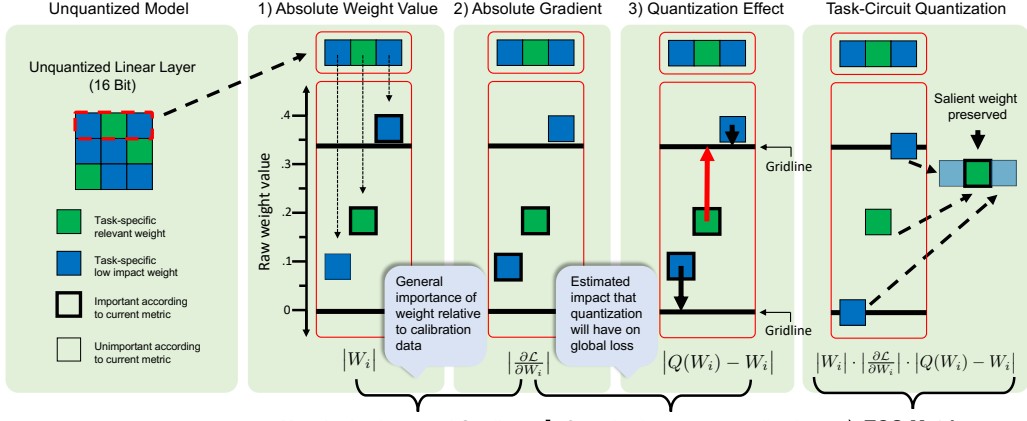

Figure 2: TACQ saliency metric is used to select relevant weights to hold out in 16-bit precision. This metric is the product of three terms which we highlight above, grouped into two factors: Magnitude-Sharpened Gradient and Quantization-Aware Localization.

methods ignore the effect of quantization (Shao et al., 2024b; Kim et al., 2024), both fail to predict changes in the global loss due to quantization. In Section 3, we introduce our approach to computing weight sensitivity and in Section 4 we compare our proposed approach with SqueezeLLM, and SPQR, two SOTA approaches that utilize outlier preservation, as well as SliM-LLM (Huang et al., 2024b), a strong mixed-precision baseline which dynamically assigns bit-width based on importance, finding our method significantly outperforms all baselines. Appendix D contains an expanded description of other mixed-precision methods and our choice of baselines.

## 3 Method

Our method is defined by a saliency metric that is used to determine important weights to preserve during quantization and consists of two parts that build on ideas from model interpretability (e.g., automatic circuit discovery, knowledge localization, and input attribution). Our metric takes two components into account, illustrated in Fig. 2:

- **Quantization-aware Localization (QAL)** traces how model performance is impacted by estimating the expected change in weights due to quantization.
- **Magnitude-sharpened Gradient (MSG)** is a generalized metric for the absolute importance of each weight by adapting methods from input attribution. This helps to stabilize TACQ and address biases caused by our use of estimations in QAL.

We combine these factors into one saliency metric that can be efficiently evaluated for every weight in one backward pass; preserving the top $p\%$ highest-scoring weights in 16 bits.

### 3.1 Quantization-Aware Localization (QAL)

A central idea in the automated circuit discovery and knowledge localization literature is comparing a corrupt model with a clean model (Meng et al., 2023; Conmy et al., 2023; Nanda, 2022). In particular, Edge-attribution patching methods (Syed et al., 2023; Kramár et al., 2024; Nanda, 2022) capture intermediate activations from a corrupt model (defined as a model with corrupted inputs) and a "clean" model with the original inputs, then use the multiplication of the difference between activations at an intermediate feature $a_i$ with its gradient $\left| a_i^{\text{clean}} - a_i^{\text{corrupt}} \right| \cdot \left| \frac{\partial \mathcal{L}}{\partial a_i^{\text{clean}}} \right|$ to attribute the importance of certain intermediate activations. We adapt this idea, but instead of applying it to activations, we apply it to weights, where we define the corrupt model as a model with the same clean input but uniformly quantized weights, and compute the gradients with respect to the weights.

Intuitively, the gradient of the final output loss with respect to the weight in the $i$th row and $j$th column of a linear layer is written as $\partial \mathcal{L}/\partial W_{ij}$ and indicates how much to move that weight in order to push the loss in a specific direction. However, another consequence of this same principle is that by multiplying a gradient by a change in weights, we obtain a linear approximation of how the loss is expected to change given the change in weights. Therefore, if we know how quantization will change weights, we can estimate which weights will cause the most damage to model performance when quantized.

To formalize this intuition, the product of the gradient and a change in weights can be understood as a first-order Taylor approximation of how much the loss will change if we were to apply a corruption or change $Q(\cdot)$ to a weight $W_{ij}$. We write $\mathcal{L}(\cdot)$ as the loss when we modify only the weight specified by the argument.

$$\mathcal{L}(Q(W_{ij})) \approx \mathcal{L}(W_{ij}) + \tfrac{\partial \mathcal{L}}{\partial W_{ij}} \cdot (Q(W_{ij}) - W_{ij}) \tag{4}$$

The second term specifies the expected change in loss and corresponds exactly to the edge attribution patching formulation when we replace activations $a$ with weights $W$.

In the context of quantization, we find the corrupt model by simulating quantization without utilizing our saliency metric, and then recording the final quantized value of each weight. We refer to this value as an estimate since extracting outliers will slightly affect the quantization process, but since the percentage of outlier weights are small (usually $\sim.35\%$) the estimate is informative. The difference between the quantized value and the original value gives us the change in weight $\left|Q(W_{ij}) - W_{ij}\right|$, where $|\cdot|$ denotes the absolute value. Consider one linear layer in a LLM, mathematically, we represent QAL as

$$\text{QAL}(W_{ij}) = \left|\tfrac{\partial \mathcal{L}}{\partial W_{ij}}\right| \cdot \left|Q(W_{ij}) - W_{ij}\right| \tag{5}$$

corresponding to terms 2 and 3 in Fig. 2.

## 3.2 Magnitude-Sharpened Gradient (MSG)

While Edge-Attribution Patching focuses on finding specific circuits through contrasting activations, input attribution methods use the product of gradient and input as a general measure of saliency for model inputs (Shrikumar et al., 2017; Ancona et al., 2019; Binder et al., 2016). This class of methods sharpens the predictive ability of the gradient by computing $\left|\tfrac{\partial \mathcal{L}}{\partial a_i}\right| \cdot |a_i|$ where $a_i$ is an scalar element of the input to the model. Note that we denote the input as $a_i$ to specify scalar input features for which the gradient of the loss is defined, which are equivalent to activations for our purposes.

Using the same strategy to replace activations with weights as in Section 3.1, we formulate the magnitude-sharpened gradient (MSG) for a specific weight $W_{ij}$ in a single linear layer as

$$\text{MSG}(W_{ij}) = |W_{ij}| \cdot \left|\tfrac{\partial \mathcal{L}}{\partial W_{ij}}\right| \tag{6}$$

corresponding to term 1 and 2 in Fig. 2. We can apply the same Taylor Approximation from equation Eq. (4) to understand the MSG by replacing $Q(W_{ij})$ with $0_{ij}$. Intuitively, this gives a generalized importance of a weight by deriving the impact of removing it entirely.[2]

MSG is crucial because it counterbalances a flaw in QAL; As QAL is weighted by $\left|Q(W_{ij}) - W_{ij}\right|$, importance for weights close to values that can be represented after quantization (i.e. values on the gridlines) is reduced to near zero, even if the gradients of these weights are large and the weights are generally important to the network. MSG ensures that a more robust general importance for weights is considered.

Our full saliency metric is the composition of QAL and MSG:

$$\text{TACQ}(W_{ij}) = |W_{ij}| \cdot \left|\tfrac{\partial \mathcal{L}}{\partial W_{ij}}\right| \cdot \left|W_{ij}^{\text{quant}} - W_{ij}\right| \tag{7}$$

---

[2]MSG falls under a class of weight importance metrics known as synaptic saliency shown to have the desirable property of behaving as circuit like synaptic flows (Tanaka et al., 2020).

Where $\left|\frac{\partial \mathcal{L}}{\partial W_{ij}}\right|$ is computed with one backward pass from a model on one calibration datapoint. We use multiple calibration datapoints by averaging of computed scores, keeping the highest-scoring $p\%$ weights in 16-bit precision and quantizing the remaining weights. We do not use MSG $\times$ QAL directly because we observe that squaring the gradient rarely matters, such as in Section 4.3, and we are only interested in order rather than the metrics' exact value. Additionally, most gradient values are $\ll 1$, and reducing such terms is beneficial for numerical precision.

### 3.3 Details on quantization implementation

TACQ uses GPTQ (Frantar et al., 2023a) for quantization, exempting salient weights identified through our metric from consideration. To obtain the gradient term of TACQ we perform backpropagation from the standard cross-entropy loss. We include in Appendix E a longer specification for our configuration of GPTQ, engineering changes needed to enable downstream conditioning, and a memory-efficient method for computing gradients. We also show that TACQ is economical in terms of quantization time in Appendix B and that we can reduce the number of examples used to compute gradients while minimally impacting performance in Appendix H.4.

## 4 Results and analysis

We base our main experimental results and analysis on the Llama-3-8B-Instruct model, with scaling experiments on Qwen2.5-32B-Instruct. We include additional results on Qwen2.5-7B-Instruct, and perplexity results on the non-instruction-tuned Llama-3-8B base model in Appendix H.

**Datasets for Quantization Conditioning and Evaluation.** To evaluate our method we pick standard datasets that cover a variety of model use scenarios. Previous work has focused on multiple choice questions and perplexity as the main evaluations. We include perplexity results in Table 14 of the appendix and focus our analysis on the accuracy of a quantized model on downstream tasks involving both multiple choice and generation. To test our method without task conditioning, we follow GPTQ to use 128 examples with sequence length 2048 for C4 (Raffel et al., 2020) and WikiText2 (Merity et al., 2016) calibration datasets.

For conditioning on downstream tasks, we use the same number of tokens as the C4 calibration dataset by sampling examples until we fill the token budget. For GSM8k (Cobbe et al., 2021) we utilize the standard 8-shot prompt and include the CoT in the conditioning dataset. We evaluate on a separate validation split. For MMLU (Hendrycks et al., 2021) we use a 5-shot prompt from the development set for each subject, drawing examples from the first 75% of examples in each subject and evaluating on the last 25% of the dataset for each subject. Due to the diversity of subjects MMLU covers, we include three MMLU splits: Humanities, Social Sciences, and STEM as defined in Hendrycks et al. (2021) to simulate more concrete tasks. We condition on them separately, effectively treating them as three independent datasets. For domain generalization experiments in Table 4, we evaluate on the Humanities split as it contains the greatest number of data points.

We include Spider (Yu et al., 2019) as a separate generation-based text-to-SQL task in a zero-shot setting, to simulate a practical application of LLMs. We draw calibration samples from the training set and evaluate on the development set using test-suite accuracy (Zhong et al., 2020). Further details are provided in Appendix F.

**Baselines and Hyperparameters** We compare against strong post-training mixed-precision quantization techniques; these generally use a saliency metric to preserve certain amounts of a model's weights in higher bit-width. Specifically, we compare against SPQR, Squeeze-LLM, and SliM-LLM. SPQR and Squeeze-LLM are the most comparable to our technique and use saliency metrics to preserve a small amount of critical weights in 16-bits. Given a target bit-width $N$, SliM-LLM uses a saliency metric to select chunks of weights to instead quantize at $N+1$ and $N-1$ bit-width.

We use the standard hyperparameters reported and base our implementation on the officially released code, adapting hyperparameter values for 2-bit and 3-bit settings. Prior methods such as SPQR and SqueezeLLM rely on threshold-based mechanisms, which make it difficult to precisely control the bit-width. As such, bit-width in tables denote the minimum bit-width observed for all runs. To ensure a fair comparison, we allow these methods to use more bits per weight. We elaborate on hyperparameter choice and calculations for average bit-width in Appendix A.

## 4.1 Main results

**2-bit – Qwen2.5-32B-Instruct**

| Method | Avg bits | MMLU Acc |
|--------|----------|----------|
| **Unquantized** | 16.00 | 82.46% |
| **GPTQ** | 2.00 | 50.55% |
| **SliM-LLM** | 2.125 | 62.51% |
| **TACQ** | **2.10** | **72.51**% |

Table 1: Application of TACQ on Qwen2.5-32B-Instruct shows that TACQ scales to larger models.

Figure 3: Comparison of accuracies (ratio) on MMLU at different compression ratios. TACQ outperforms baselines at all budgets.

We report our main results for Llama-3-8B-Instruct in Tables 2 and 3. TACQ shows major improvements over existing SOTA approaches and the GPTQ baseline. In the 2-bit setting, TACQ surpasses SliM-LLM by absolute margins of 16.0% (20.1% to 36.1%) on GSM8k, 14.1% (34.8% to 49.2%) on MMLU, and 21.9% (0% to 21.9%) on Spider, while other baselines degrade to near-random performance. In the 3-bit scenario, TACQ preserves approximately 91%, 96%, and 89% of the unquantized accuracy on GSM8k, MMLU, and Spider, respectively. Moreover, it outperforms the strongest baseline, SliM-LLM, by 1–2% across most datasets. Additionally, we show that TACQ generalizes to larger models by quantizing the Qwen2.5-32B-Instruct model, presenting our results in Table 1. TACQ recovers 87.93% of the 16-bit unquantized model's MMLU performance at 2-bit, reducing the model size by around 8x and outperforming SliM-LLM by 10.00%.

In Fig. 3, we demonstrate the trade-off between accuracy and average bit-width for different approaches, showing TACQ consistently outperforms. We describe the hyperparameters tuned for this plot in Appendix A. Additionally, we include a comparison with SPQR in an apples-to-apples setting due to its larger hyperparameter space in Appendix G.

For generation settings where the model must output multiple tokens sequentially, our method represents a qualitative change in the ability of the quantized model. In the Spider task, TACQ is the only method that recovers non-negligible performance (above zero) in the 2-bit setting. We include qualitative examples in Appendix C for both Spider and GSM8k, where other methods produce random tokens or lose their instruction following capability.

We observe that quantization affects the model accuracy more drastically in the GSM8k and Spider settings where the model must generate tokens sequentially. We theorize that this could be due to the compounding of errors from quantization during generation.

## 4.2 Task-conditioning transfer

Existing methods assume that quantization performance is relatively unaffected by the choice of calibration set (Williams & Aletras, 2024; Paglieri et al., 2024), and we show in

**2-bit**

| Method | Avg bits | GSM8k | MMLU Splits | | | |
|--------|----------|-------|-----------------|------|------------|------|
| | | | Social Sciences | STEM | Humanities | Full |
| Full | 16 | 73.46 | 75.74 | 54.88 | 63.42 | 66.51 |
| GPTQ | 2.00 | $2.86 \pm 0.49$ | $26.06 \pm 1.44$ | $27.84 \pm 1.19$ | $26.08 \pm 1.45$ | $26.76 \pm 0.17$ |
| SliM | 2.125 | $20.12 \pm 1.31$ | $40.99 \pm 0.86$ | $32.94 \pm 1.27$ | $35.31 \pm 1.25$ | $34.84 \pm 3.92$ |
| Squeeze | 2.15 | $2.07 \pm 0.50$ | $26.02 \pm 1.81$ | $27.92 \pm 1.78$ | $26.62 \pm 0.80$ | $26.27 \pm 0.87$ |
| SPQR | 2.14 | $1.24 \pm 0.16$ | $25.85 \pm 1.87$ | $23.88 \pm 1.03$ | $23.77 \pm 0.85$ | $25.90 \pm 0.99$ |
| TACQ | 2.10 | $\mathbf{36.11 \pm 0.49}$ | $\mathbf{56.34 \pm 1.89}$ | $\mathbf{41.12 \pm 0.27}$ | $\mathbf{47.98 \pm 0.91}$ | $\mathbf{49.19 \pm 0.37}$ |

**3-bit**

| Method | Avg bits | GSM8k | MMLU Splits | | | |
|--------|----------|-------|-----------------|------|------------|------|
| | | | Social Sciences | STEM | Humanities | Full |
| Full | 16 | 73.46 | 75.74 | 54.88 | 63.42 | 66.51 |
| GPTQ | 3.00 | $51.96 \pm 0.52$ | $64.17 \pm 0.97$ | $47.14 \pm 1.32$ | $51.20 \pm 1.15$ | $56.30 \pm 0.48$ |
| SliM | 3.125 | $66.24 \pm 2.23$ | $71.40 \pm 0.20$ | $51.36 \pm 1.23$ | $58.54 \pm 0.81$ | $61.95 \pm 0.03$ |
| Squeeze | 3.15 | $49.99 \pm 4.04$ | $69.25 \pm 2.71$ | $51.14 \pm 0.85$ | $57.01 \pm 1.99$ | $59.66 \pm 0.37$ |
| SPQR | 3.14 | $54.99 \pm 1.54$ | $66.15 \pm 0.84$ | $49.78 \pm 1.35$ | $56.17 \pm 1.03$ | $58.65 \pm 0.61$ |
| TACQ | 3.12 | $\mathbf{67.07 \pm 1.55}$ | $\mathbf{71.66 \pm 0.60}$ | $\mathbf{52.42 \pm 0.77}$ | $\mathbf{60.54 \pm 1.85}$ | $\mathbf{63.28 \pm 0.41}$ |

Table 2: Comparison of downstream task accuracy (%) across quantization methods on Llama-3-8B-Instruct. We report the three-run average results on both 2-bit and 3-bit settings utilizing different calibration dataset seeds.

| Method | 2-bit | | | | | 3-bit | | | | |
|--------|-------|------|---------|------|------|-------|-------|---------|-------|-------|
| | GPTQ | SliM | Squeeze | SPQR | TACQ | GPTQ | SliM | Squeeze | SPQR | TACQ |
| **Avg Bits** | 2.00 | 2.125 | 2.15 | 2.14 | **2.12** | 3.00 | 3.125 | 2.15 | 3.14 | **3.12** |
| **Spider Acc** | 0.15 | 0.00 | 0.00 | 0.00 | **21.92** | 36.65 | 58.32 | 46.33 | 46.16 | **60.41** |

Table 3: Spider accuracy (%) for 2- and 3-bit settings. Unquantized performance is 67.6%.

Appendix E.5 that most methods are implemented such that conditioning on downstream datasets is difficult. We present evidence that calibration datasets have a large impact in 2- to 3-bit settings in Table 4 and Table 5. When conditioning on WikiText2, a standard calibration dataset, SliM-LLM fails to produce any result on MMLU Humanities and reduces to random performance in 2-bit, but conditioning consistently raises performance by more than 10 percentage points. Analyzing the impact for GPTQ at a broader range of bit-widths shows that the effect of conditioning is much smaller at 4 bits and above, and that testing on generation settings can widen the gap.

In Table 4, we present evaluation results on the MMLU Humanities validation split under different task-conditioning settings for TACQ, SliM-LLM, and GPTQ. Compared to using a general dataset like WikiText2 for conditioning, leveraging the MMLU Humanities calibration set yields a 10–17% improvement in the 2-bit setting and a 3–5% gain in the 3-bit setting for both TACQ and SliM-LLM. Furthermore, using a calibration dataset with a closer distribution, such as the MMLU STEM split, can also improve results by approximately 6% in the 2-bit setting and 3% in the 3-bit setting for TACQ and SliM-LLM. We observe a similar trend for GPTQ in the 3-bit setting, while performance stays random in the 2-bit case.

## 4.3 Saliency metrics and ablations

We compare several saliency scores in this section, where saliency is defined based on the following criteria: (1) Weight: the magnitude of the weight; (2) Sample Fisher: the diagonal

| | MMLU Humanities Accuracy | | | | | |
|---|---|---|---|---|---|---|
| **Avg Bits** | **2-bit** | | | **3-bit** | | |
| **Method →** **Conditioned On ↓** | GPTQ | SliM-LLM | TACQ | GPTQ | SliM-LLM | TACQ |
| Humanities | 26.08 | 35.31 | **47.98** | 51.20 | 58.54 | **60.23** |
| STEM | 25.06 | 29.97 | **37.60** | 44.88 | 56.99 | **59.86** |
| WikiText2 | 24.72 | 23.96 | **31.16** | 41.66 | 53.60 | **56.48** |

Table 4: Comparison of MMLU Humanities accuracies (%) when conditioned on MMLU Humanities, MMLU STEM, or WikiText2. We examine our method, the strongest baseline SliM-LLM, and GPTQ. *Our method performs better than the strongest baseline regardless of the conditioning dataset used by 8-10 points consistently in 2-bits and 2-3 points in 3-bits.* We also show that conditioning has high impact for all methods in the 2- to 3-bit settings.

| MMLU Accuracy | | | | | GSM8k Accuracy | | | | |
|---|---|---|---|---|---|---|---|---|---|
| **Conditioning** | **2bit** | **3bit** | **4bit** | **8bit** | **Conditioning** | **2bit** | **3bit** | **4bit** | **8bit** |
| MMLU_MCQA | 26.7 | 56.3 | 63.0 | 66.6 | GSM8k | 2.9 | 52.0 | 65.0 | 72.6 |
| C4 | 24.6 | 45.7 | 62.3 | 66.5 | C4 | 2.0 | 17.5 | 62.6 | 72.5 |

Table 5: MMLU accuracy (%) for GPTQ quantization of Llama-3-8B-Instruct conditioned on the downstream dataset vs the general C4 dataset.

of the empirical Fisher information matrix as used in Kim et al. (2024); (3) Sample Gradient: the sum of per-sample gradients; (4) Sample Absolute Gradient: the sum of the absolute values of per-sample gradients. We include the mathematical formulation to compute each metric on one conditioning sample.

Note that in this comparison, only the saliency score varies. The rest of the pipeline, including top-$p\%$ selection and quantization with GPTQ, remains unchanged as described in Section 3. Table 6 presents the results on MMLU under 2-bit and 3-bit settings. We observe that Weight performs the worst, likely due to its lack of both task-specific and global information. Sample Fisher, as in (Kim et al., 2024), and Sample Absolute Gradient, as in (Shao et al., 2024b), produce similar results, this is

| Sensitivity Measure | 2-bits | 3-bits |
|---|---|---|
| Weight: $\|W_{ij}\|$ | 32.98 | 59.79 |
| Sample Fisher: $(\partial\mathcal{L}/\partial W_{ij})^2$ | 41.80 | 62.20 |
| Sample Gradient: $\partial\mathcal{L}/\partial W_{ij}$ | 34.23 | 61.74 |
| Sample Absolute Gradient: $\|\partial\mathcal{L}/\partial W_{ij}\|$ | 42.70 | 62.09 |
| MSG: $\|W_{ij}\| \cdot \|\partial\mathcal{L}/\partial W_{ij}\|$ | 47.86 | 63.25 |
| QAL: $\|\partial\mathcal{L}/\partial W_{ij}\| \cdot \|W_{ij}^{\text{quant}} - W_{ij}\|$ | 47.75 | 63.22 |
| TACQ: $\|W_{ij}\| \cdot \|\partial\mathcal{L}/\partial W_{ij}\| \cdot \|W_{ij}^{\text{quant}} - W_{ij}\|$ | **49.19** | **63.90** |

Table 6: Ablations on sensitivity metrics, evaluated on MMLU Accuracy (%).

likely due to the fact that the Sample Fisher is approximated as the squared of the gradient in Kim et al. (2024). Gradient underperforms most of the other approaches because its values can cancel out across samples due to not taking the absolute value.

Recall that our final metric for TACQ is $\|W_{ij}\| \cdot \left|\frac{\partial\mathcal{L}}{\partial W_{ij}}\right| \cdot \left|W_{ij}^{\text{quant}} - W_{ij}\right|$. We consider the QAL and MSG components individually and show that each improves performance, and when combined to form the TACQ metric the performance improves further. Augmenting Sample Absolute Gradient with the first term or the third term gives us MSG and QAL respectively, both individually improving 2-bit performance by approximately 5%. Combining all three components achieves the best results in both 2-bit and 3-bit scenarios.

## 5    Conclusion

Building on insights from mechanistic interpretability, we propose TACQ, a task-aware method for post-training quantization that substantially improves model performance at ultra-low bit-widths (2- to 3-bits), a regime where prior methods often degrade to near-random outputs. By selectively preserving only a small fraction of salient weights in 16-bit precision, our approach aligns with work in automatic circuit discovery finding that sparse weight "circuits" carry disproportionately large responsibility for specific tasks. This leads to substantial gains in downstream evaluation, such as nearly doubling GSM8k scores in the 2-bit setting, while maintaining the memory savings that make local and embedded deployments feasible. Moreover, TACQ also translates to general-purpose settings, where it also beats mixed-precision baselines, and outperforms baselines across memory budgets. Going beyond multiple-choice settings, our experiments on Spider show that TACQ better preserves the model's generation ability, making our method applicable to program-prediction tasks. This would apply also to agentic settings, where models predict often large numbers of executable outputs and where efficiency is a concern.

## Acknowledgments

We thank Prateek Yadav and Duy Nguyen for their helpful comments and feedback on this paper. This work was supported by NSF-CAREER Award 1846185, DARPA ECOLE Program No. HR00112390060, and NSF-AI Engage Institute DRL-2112635. Any opinions, findings, and conclusions or recommendations in this work are those of the author(s) and do not necessarily reflect the views of the sponsors.

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

## A   Configurations and Hyperparameter Tuning

We describe the hyperparameters we tune for various baselines and our method and briefly describe some engineering details when relevant.

**SPQR (Dettmers et al., 2023).**   We utilize the default configuration in the official implementation: groupsize 16, quantization statistics in 3-bits, and double quantization groupsize of 16. SPQR has 6 adjustable hyperparameters, which gives SPQR a great flexibility in terms of being able to make different tradeoffs. We consider a standard setting in the main tables and include a pareto plot on an apples to apples setting in Appendix G. Specifically for the main tables, for the 3 bit setting we use base bit-width 2, groupsize 16, quantization statistics in 3-bits and double quantization groupsize of 16. For the 2 bit setting we use base bit-width 1, groupsize 16, quantization statistics in 3-bits and double quantization groupsize of 16. The small groupsize of SPQR means that to match bit-width with other methods, it is often compared such that it uses $N - 1$ as the base bit-width, where $N$ is the target bit-width (Dettmers et al., 2023; Kim et al., 2024). The number of outliers is variable due to our use of calibration dataset seeds, conditioning datasets, and SPQR's usage of a threshold. Specifically, the percentage of outliers depends on the calibration dataset seed, model, and bit-width. Critically, as it depends on the conditioning dataset seed, bit-width varies with every run in our main tables. In practice, we observe SPQR to use around 1-2% outliers and we only consider quantization runs where SPQR uses higher average bit-width than our method, noting the lowest bit-width in the bit-width sections of tables. We note that in the non-task conditioned setting with one seed, the thresholding problem is not as big of an issue, as we only need to quantize the model once. Variability arises since we need to condition on each seed and downstream dataset.

For the Pareto plot in Appendix G, we tune the threshold value to increase and decrease the percentage of outliers preserved.

**SqueezeLLM (Kim et al., 2024).**   SqueezeLLM uses a threshold to produce the percentage of weight outliers, due to this, the percentage of outliers vary slightly between different models, and we specify the percentages used below. For both Llama-3-8B-Instruct and Qwen2.5-7B-Instruct we use SqueezeLLM with .45% weight outliers and .05% Fisher-based outliers. We calculate the bit-width as bit-width $= N - Nr + 32r$ where $r$ is the ratio of outliers we use, this is the same formula used for our method. We use .46% weight outliers and .05% Fisher based outliers for the Llama-3-8B base model. For SqueezeLLM we use a number of outliers that is greater than the number of outliers used for TACQ, and we also validate that TACQ uses a average bit-width and percentage of outliers that is lower than the .4% weight outliers and .05% Fisher-based outliers used in the SqueezeLLM paper (Kim et al., 2024).

For the Pareto plot Fig. 3, we tune the percentage of Fisher-based outliers preserved.

**SliM-LLM (Huang et al., 2024b).**   We utilize the default groupsize of 128 described to be the optimal in the paper Huang et al. (2024b).

For the Pareto plot Fig. 3, we follow the paper to adjust the groupsize. However, we found that decreasing the groupsize increases quantization time. In Table 7 we show the quantization time for SliM-LLM on a NVIDIA RTX A6000 GPU with 48GB of RAM for Llama-3-8B-Instruct in the 2-bit setting, where we observe that the time taken for quantization scales rapidly with decreased groupsize. Due to resource constraints, we do not evaluate

| Configuration | SliM-LLM Quantization Time |
|---|---|
| g512 | 1 hour 58 minutes 14.209 seconds |
| g256 | 9 hours 26 minutes 5.671 seconds |
| g64 | 4 days 1 hour 40 minutes 0.648 seconds |

Table 7: Runtime for each SliM-LLM baseline configuration

SliM-LLM beyond groupsize 64, which results in a increase of .25 average bit-width on top of the base bit-width. We calculate the average bit-width using a conservative formula of bit-width $= N + 16/g$ where $g$ is the groupsize assuming one 16-bit scale per group and $N$ is the base bit-width.

**GPTQ (Frantar et al., 2023a).** We compare against GPTQ with channel-wise quantization and MSE based determination of scales, which is what TACQ uses as a baseline quantization method.

**TACQ.** TACQ only has one hyperparameter that uniquely determines the bit-width. We calculate bit-width as bit-width $= N - Nr + 32r$ where $r$ is the ratio of outliers we preserve. For TACQ quantization on NVIDIA RTX A6000 GPUs with 48GB of RAM, it takes around 2.76 hours to capture gradients and .58 hours to quantize the model with the GPTQ step for Qwen2.5-7B-Instruct, where our current implementation for the capturing of gradients requires 93.29GB of VRAM at peak usage. For Qwen2.5-32B-Instruct, it takes around 4.37 hours to capture gradients and 2.56 hours to quantize the model with the GPTQ step on 138.62GB of RAM with NVIDIA RTX 6000 Ada Generation GPUs. We note that with further engineering optimizations, such as gradient checkpointing, memory overhead can be reduced, but we found the current implementation sufficient for our experiments due to the relatively short time required for gradient capture and discuss it in Appendix E. In particular, Appendix E.3 describes the engineering details used to obtain compute gradients as part of TACQ.

For the Pareto plot Fig. 3, we tune the percentage of outliers preserved.

## B  Quantization Time and Compute Cost

While gradient capture is expensive, there are several reasons why we believe it is tractable in the context of TACQ.

- Gradients only need to be captured once to freely quantize at varied % outliers and base bitness.

- Compared to the quantization timelines and overheads of other methods, TACQ has reasonable and sometimes superior quantization timeline, see Table 8.

- The number of datapoints used for gradient capture can be reduced by 8x without affecting performance. See Table 15.

| Method | Compute Time (hours) | GPU Memory Peak (GB) |
|---|---|---|
| GPTQ 7B | 0.58 | 7.68 |
| SliMLLM 7B | 14.66 | 12.48 |
| TACQ 7B | 2.76 + 0.58 | 93.29 |
| GPTQ 32B | 2.56 | 20.16 |
| SliMLLM 32B | 54.71 | 22.608 |
| TACQ 32B | 4.37 + 2.56 | 138.62 |

Table 8: Compute time and peak GPU memory for different quantization methods

# C   Qualitative Results

**Question:** Janet's ducks lay 16 eggs per day. She eats three for breakfast every morning and bakes muffins for her friends every day with four. She sells the remainder at the farmers' market daily for $2 per fresh duck egg. How much in dollars does she make every day at the farmers' market? **Answer:** 18

| Method | Example Output | Outcome |
|---|---|---|
| GPTQ | *'Janet's ducks laid 16 eggs per day. She eats three for breakfast every day. She sells the remainder at the farmers' market for $2 per duck every day. She makes 2 x 1 = 2 dollars per day. The answer is 2."* | Failure |
| SPQR | *'egal the Ramplib-Jetonunga send Index, Bryantonders(TM)[continues...]* | Failure |
| Squeeze | *'Janet started with 16 eggs per day. She eats 3 for breakfast, then bakes 4 for her friends daily. 2, 2, 4 = 6. The answer is 6."* | Failure |
| SliM | *'She eats 3 muffins for breakfast. She bakes 4 muffins every day. 3 + 4 is 7. That means she bakes 7 muffins per day. If she sells 16 - 7 = 9 eggs for 2 dollars each, then she makes 9 x 2 = 18. The answer is 18.#### 1. What is the value of the variable you use?A: The answer is 18. The answer is 18. The answer is 18.[repeats...] "* | Success |
| TACQ | *'Janet lays 16 eggs per day. She eats 3 eggs for breakfast, so 16 - 3 = 13 eggs are left. After baking 4 eggs for muffins, 13 - 4 = 9 eggs are left. She sells 9 eggs for 2 dollars per egg, so 9 x 2 = 18. The answer is 18."* | Success |

Table 9: Comparison of GSM8k CoT output in 2-bit quantization with Llama-3-8B-Instruct.

**Question:** What are all distinct countries where singers above age 20 are from? **Answer:** SELECT DISTINCT country FROM singer WHERE age > 20

| Method | Example Output | Outcome |
|---|---|---|
| GPTQ | *"';* | Failure |
| SliM | *### What are the best ways to win, and what are the best ways to, to ride, and to ride, and to ride, and to ride[repeats...]* | Failure |
| SPQR | *achts(disposing AIGN didSelectRowAtIndexPath fymalavek#Regionanceeryzer Spy[continues...]* | Failure |
| Squeeze | *### 1.1.1.1.1.1.1.1.1.1.1.1.1.1.1.1.1.1.1.1.1.1.1.1.1.1.1.1.1.1.1.1.1.1.1.[repeats...]* | Failure |
| TACQ | *SELECT DISTINCT Country FROM singer WHERE Age > 20;* | Success |

Table 10: Comparison of method outputs at 2-bit quantization with Llama-3-8B-Instruct.

In Table 9 we show GSM8k CoT generations from all methods in a 2-bit setting for Llama-3-8B-Instruct. We note that while SPQR Dettmers et al. (2023) is the only method not to generate text resembling a response, this is potentially due to the fact that we evaluate SPQR with base-bitwidth of 1 in order to match the approximately 2-bit setting. With higher base bit-width and a different choice of hyperparameters it should be possible for SPQR to achieve better results, for this reason we include the apples to apples setting in Appendix G to marginalize the impact of hyperparameters. In Table 10, we show Spider generation results from all methods in a 2-bit setting for Llama-3-8B-Instruct. TACQ is the only method to recover task relevant results in 2-bits for Llama-3-8B-Instruct. The next-best method, SliM-LLM, loses instruction following ability, but it does produce legible text. The examples in Table 10 are generally representative of the type of output for each method. For TACQ we observe that all results look like SQL queries, but not all queries are syntactically or semantically correct. For other methods, the majority of generations do not resemble SQL queries.

# D   Expanded Background

We replicate some of the information found in our background section and include this expansion in our notation for a more comprehensive discussion.

**GPTQ and Compensation in Quantization.** Many methods frame the quantization problem as layerwise reconstruction, seeking to minimize layer-wise reconstruction loss after

quantization (Frantar et al., 2023a; Huang et al., 2024b; Dettmers et al., 2023). Our method builds on GPTQ quantization, which makes independent adjustments to the values of unquantized weights in each row to compensate for error induced by quantizing each additional column. Let $L$ denote the error function, which can be expanded using a Taylor series as

$$\delta L = (\frac{\delta L}{\delta w})^T \delta w + \frac{1}{2}\delta w^T H \delta w + O(||\delta w||^3) \tag{8}$$

where $H = \delta^2 L / \delta w^2$ is the Hessian matrix. In the layer-wise quantization scenario where $L = ||WX - Q(W)X||_2^2$ (i.e., the layer-wise reconstruction loss), the Hessian simplifies to $H = XX^T$. Note that, we can neglect the first-order terms if the model is trained to convergence, as well as the terms higher than the second-order. GPTQ aims to minimize the loss increase induced by quantizing a weight $w_q$:

$$\min_{q, \delta w} \frac{1}{2}\delta w^T H \delta w, \text{ s.t. } e_q^T \delta w + w_q = Q(w_q) \tag{9}$$

where $e_q^T$ is the $q$-th canonical basis vector. To solve this constrained optimization problem, we can use the method of Lagrange multipliers to reformulate the original loss function with the condition, leading to the Lagrangian $\frac{1}{2}\delta w^T H \delta w + \lambda(e_q^T \delta w + w_q - Q(w_q))$. Setting its derivative to zero yields the optimal solution to $\delta w$ and the quantization order of row index $q$:

$$q^* = \text{argmin}_q \frac{(Q(w_q) - w_p)^2}{[H_F^{-1}]_{qq}}, \quad dw = -\frac{w_q - Q(w_q)}{[H_F^{-1}]_{qq}} \cdot (H_F^{-1})_{:,q}. \tag{10}$$

where $F$ denotes the set of unquantized (full-precision) weights. In practice, we quantize the weight in an arbitrary order rather than using the greedy order derived in the first term in Eq. (10), which simplifies the process and empirically demonstrates similar quantization performance in Frantar et al. (2023a). After each quantization step, the remaining weights are adjusted according to the second term in Eq. (10) to compensate for quantization-induced error.

**Other Post-Training Quantization Techniques.** Weight-quantization methods can be broadly categorized into quantization-aware training (QAT) (Ma et al., 2024; Liu et al., 2023a) and post-training quantization (PTQ). QAT achieves higher accuracy by simulating the quantization process during the forward pass and optimizing weights over large datasets. However, it introduces significant computational overhead, making it less practical in many scenarios. In contrast, PTQ has emerged as an effective alternative, enabling model compression using a small calibration dataset without expensive retraining. For example, OBC (Frantar et al., 2023b) and GPTQ (Frantar et al., 2023a) efficiently quantize weight columns using locally approximated second-order gradient information. AWQ (Lin et al., 2024) reduces the quantization errors by theorizing weights adjacent to large input activations are important and scaling to protect them. OmniQuant (Shao et al., 2024a), AQLM (Egiazarian et al., 2024), and PV-Tuning (Malinovskii et al., 2024) improve quantization performance by training the quantization parameters. TACQ builds upon GPTQ; however our proposed saliency metric are not restricted to this framework and can be applied more broadly.

**Other Mixed-Precision Quantization Techniques.** To achieve higher compression ratios, mixed-precision methods exploit the fact that not all weights of an LLM contribute equally to performance. Instead of uniformly quantizing all weights to a fixed precision, these methods allocate different bit precision to weights based on weight importance, preserving

critical weights in higher precision while compressing less important ones more aggressively. While this approach can slightly increase inference time compared to uniform precision quantization, it often leads to better overall model performance. Weight importance in LLMs tends to follow a power-law distribution, meaning that preserving even a small fraction of high-importance weights (e.g., 1%) can drastically improve performance (Dettmers et al., 2023; Yu et al., 2024). Since it is possible to monotonically improve model performance by preserving more weights, this type of mixed-precision method can be added on top of many existing approaches to further improve performance. For example, SPQR (Dettmers et al., 2023) uses it in combination with double quantization, TACQ uses it with channel-wise GPTQ, and SqueezeLLM (Kim et al., 2024) uses it with k-means based quantization.

SPQR (Dettmers et al., 2023) and SqueezeLLM (Kim et al., 2024) identify and maintain $< 1\%$ of weights in 16-bit to push performance in 3-bit and 4-bit settings. However, SPQR makes use of GPTQ's local layerwise metric to estimate important weights, while SqueezeLLM uses weight magnitude to identify the majority of its outliers, consisting of .4% of total weights, and uses Fisher information to identify the remaining .05%. SliM-LLM also uses the GPTQ metric to estimate weight-importance but allows a more flexible assignment of weights by pushing columns groups in the weight matrices to $N$-1 bit or $N$+1 bit alternatively where $N$ is the desired bit-width, achieving higher performance. These above methods also introduce techniques including small group sizes (SPQR, SliM-LLM) double quantization (SPQR), and K-Means clustering (SqueezeLLM) to achieve their performance gains. Mixed-precision has been used to compress models to 1 bit per weight, but binarization techniques often take different approaches to quantization due to the extreme compression ratio and many require training to be effective. In terms of training free binarization methods such as PB-LLM (Shang et al., 2023) and BiLLM (Huang et al., 2024a), SliM-LLM exceeds these techniques when considered in 2- and 3-bit settings.

Our method is the first to leverage both the predicted effects of quantization and global loss information to identify outliers, consistently outperforming these methods by more than 20% in 2-bits and more than 5% in 3-bits. Additionally, we do not use a threshold and therefore does not need to employ any search to achieve desired outlier ratios.

**Not all parts of an LLM are equally important**  Not all weights in a given LLM are equally important for a given task, and information density in LLMs is low, as indicated by the fact that parts of models can be removed outright and small subnetworks can be trained to recover the original LLM's performance (Ma et al., 2023; Men et al., 2024). Furthermore, research attempting to provide a mechanistic understanding of LLMs by studying activation circuits has revealed evidence of task-specific weights in LLMs, where small subsets of weights in an LLM are especially important for specific tasks (Christ et al., 2024; Olah et al., 2020; Kramár et al., 2024; Syed et al., 2023; Dai et al., 2022). Not only certain weights are important in networks, different tokens also have variable importance and some work use this orthogonal observation to compress LLMs (Zhang et al., 2023; Liu et al., 2023b), a concurrent work uses token importance information to adjust the objective function for weight quantization (Sung et al., 2025).

# E  Expanded Description of TACQ

We augment the description of our method in this appendix with an expanded discussion.

Our method is defined by a saliency metric – used to determine important weights to preserve during quantization – and consists of two parts building on ideas from model interpretability (e.g., automatic circuit discovery, knowledge localization, and input attribution). Our metric takes two components into account, illustrated in Fig. 2:

- **Quantization-aware Localization (QAL)** traces how model performance is impacted by changes to the weight by estimating the expected change due to quantization.
- The **Magnitude-sharpened Gradient (MSG)** is a generalized metric for the absolute importance of each weight by adapting methods from input attribution. This helps to stabilize TACQ and address biases caused by our use of estimations in QAL.

We combine these factors into one saliency metric that can be efficiently evaluated for every weight in one backward pass; we then preserve the top $p\%$ highest-scoring weights in 16-bits.

### E.1 Quantization-Aware Localization (QAL)

A central idea in the automated circuit discovery and knowledge localization literature is comparing a corrupt model with a clean model (Meng et al., 2023; Conmy et al., 2023; Nanda, 2022). Intermediate activations in the corrupt model (defined as a model with corrupted inputs) are compared to a "clean" model with the original inputs. By replacing certain clean activations with corrupt activations, critical pathways in the model can be identified by measuring which activations, when perturbed, most affect the output. Edge-Attribution Patching methods (Syed et al., 2023; Kramár et al., 2024; Nanda, 2022) further use the gradient and the difference between activations in order to estimate the impact on final model output, allowing attribution scores to be computed efficiently. The attribution to a particular intermediate activation is defined in two concise terms: $\left|\frac{\partial \mathcal{L}}{\partial a_i}\right| \cdot \left|a_i^{\text{clean}} - a_i^{\text{corrupt}}\right|$. We borrow this idea, but instead of applying it to activations, we apply it to weights, where we define the corrupt model as a model with the same clean input but corrupted weights, and compute the gradients with respect to the weights instead of activations.

Specifically, the gradient of the final output loss with respect to the weight in the $i$th row and $j$th column of a linear layer, written as $\partial \mathcal{L}/\partial W_{ij}$, indicates how much to move that weight in order to push the loss in a specific direction. However, another consequence of this same principle is that by multiplying a gradient by a change in weights, we obtain a linear approximation of how the loss is expected to change given the change in weights. Therefore, if we know how quantization will change weights, we can estimate which weights will cause the most damage to model performance when quantized.

To formalize this intuition, the product of the gradient and a change in weights can be understood as a first-order Taylor approximation of how much the loss will change if we were to apply a corruption or change $Q(\cdot)$ to a weight $W_{ij}$. We write $\mathcal{L}(\cdot)$ as the loss when we modify only the weight specified by argument.

$$\mathcal{L}(Q(W_{ij})) \approx \mathcal{L}(W_{ij}) + \frac{\partial \mathcal{L}}{\partial W_{ij}} \cdot (Q(W_{ij}) - W_{ij}) \tag{11}$$

We are interested in the second term, which specifies the expected change in loss. Note that this term corresponds exactly to the edge attribution patching formulation when we replace activations $a$ with weights $W$.

In the context of quantization, we find the corrupt model by simulating the quantization all weights without utilizing our saliency metric and then recording the final quantized value of each weight. We refer to this value as an estimate since extracting outliers will slightly affect the quantization process, but since the percentage of outlier weights are small (.35%) the estimate is informative. The difference between the quantized value and the original value gives us the change in weight $\left|Q(W_{ij}) - W_{ij}\right|$, where $|\cdot|$ denotes the absolute value. Consider our channel-wise quantization setting at 2-bits, where each *channel* is a row in a weight matrix, and is associated with four possible *gridlines*, which are values that can be represented in the final quantized format. The expected change then reveals how near or far the weight lies from a quantization gridline. In a 2-bit scenario, there are only four gridlines per channel. Consider one linear layer in a LLM, mathematically, we represent QAL as

$$\text{QAL}(W_{ij}) = \left|\frac{\partial \mathcal{L}}{\partial W_{ij}}\right| \cdot \left|Q(W_{ij}) - W_{ij}\right| \tag{12}$$

corresponding to terms 2 and 3 in Fig. 1.

### E.2 Magnitude-Sharpened Gradient (MSG)

Input attribution methods for models trained with gradient descent have used the product of gradient and input as a measure of saliency (Shrikumar et al., 2017; Ancona et al., 2019), as this sharpens the predictive ability of the gradient. Shrikumar et al. (2017) also show that

this is equivalent to layerwise relevance propagation methods (Binder et al., 2016). This class of methods compute $\left|\frac{\partial \mathcal{L}}{\partial a_i}\right| \cdot |a_i|$ where $a_i$ is an scalar element of the input to the model. Note that we denote the input as $a_i$ to specify scalar input features for which the gradient of the loss is defined, which are equivalent to activations for our purposes.

Utilizing the same strategy to replace activations with weights as in Appendix E.1 we formulate the magnitude-sharpened gradient (MSG) for a specific weight $W_{ij}$ in a single linear layer as

$$\text{MSG}(W_{ij}) \ = \ |W_{ij}| \ \cdot \ \left|\frac{\partial \mathcal{L}}{\partial W_{ij}}\right| \tag{13}$$

corresponding to term 1 and 2 in Fig. 1. We can apply the same Taylor Approximation from equation Eq. (11) to understand MSG by replacing $Q(W_{ij})$ with $0_{ij}$. Intuitively, this gives a generalized importance of a weight by asking what would be the effect if we were to remove it entirely. MSG falls under a class of weight importance metrics known as synaptic saliency shown to have the desirable property of behaving as circuit like synaptic flows (Tanaka et al., 2020).

Apart from acting as a general synaptic importance for weights, MSG counterbalances a flaw in QAL. Specifically, because QAL is weighted by $\left|Q(W_{ij}) - W_{ij}\right|$, importance for weights close to values that can be represented after quantization (i.e. values on the gridline) is reduced to near zero, even if the gradient of these weight is large and the weights are generally important to the network. Recall that QAL does not have a perfect approximation of $\left|Q(W_{ij}) - W_{ij}\right|$, then minimizing importance in these weight would be detrimental. Thus, MSG serves as a critical counterbalancing factor such that a more robust general importance of a weight is considered.

Additionally, 0 is always a quantization gridline and the majority of weights lie within a small range around zero in a unimodal distribution (Kim et al., 2024), therefore this term is a good approximation of the expected change in value of many weights and does not conflict with our earlier quantization aware localization (Ancona et al., 2019; Shrikumar et al., 2017).

Our full saliency metric is the composition of QAL and MSG:

$$\text{TACQ}(W_{ij}) \ = \ |W_{ij}| \ \cdot \ \left|\frac{\partial \mathcal{L}}{\partial W_{ij}}\right| \ \cdot \ \left|W_{ij}^{\text{quant}} - W_{ij}\right| \tag{6}$$

Where $\left|\frac{\partial \mathcal{L}}{\partial W_{ij}}\right|$ is computed with one backward pass from a model from one calibration datapoint. We combine multiple calibration datapoints by taking the average of computed scores, keeping the highest-scoring $p\%$ weights in 16-bit precision and quantizing the remaining weights.

Concretely, we expect that when QAL is used, the average distance to gridlines will increase. Intuitively, given the same gradient importance, it is more valuable to preserve outliers far away from gridlines as they will be more heavily impacted when quantized. We compute this average distance metric and show that this is indeed the case.

| Scale | Gradient | QAL |
|---|---|---|
| Outlier distance to gridlines | 3.426e-3 ± 1.604e-3 | 9.577e-3 ± 4.643e-3 |

Table 11: Average distance to gridlines for preserved weights under QAL vs gradient selection strategies.

### E.3   Computing the Gradient Efficiently.

Our primary goal is to compute the loss gradient with respect to the weights without incurring the memory costs associated with training. We note that training usually incurs an large cost when using modern methods. When training with an optimizer based on Adam (Kingma & Ba, 2017), we must store the gradient, the model weights, and the first and second moments—leading to large memory requirements. Each of these components require as much memory to store as the model weights themselves. Since we only collect

gradients there is no need to store these moment parameters, reducing memory costs from 4x model size to 2x model size.

To reduce memory usage from 2x model size to 1x model size we take advantage of decomposing the gradient with respect to weights from gradients with respect to intermediate activations. The gradient with respect to the weights $\partial \mathcal{L}/\partial W_{ij}$ can be computed directly from the input activation and gradient output matrices. Given a batch size of one, let the vector of input activations to a linear layer be $a$ and the output activations of the same layer prior to applying a nonlinearity be $z$:

$$\frac{\partial L}{\partial W_{ij}} = a_i^{(l-1)} \frac{\partial L}{\partial z_j}$$

When batch size $> 1$, $z$ is expanded to become matrices.

Consequently, it is not necessary to keep the entire model's gradients in GPU memory. Instead, we cache the activations and the output gradients on the CPU and disable gradient computation for the weights during the backward pass while still allowing the gradients to be computed for activations. We calculate the final weight gradients on the CPU based on the cached activations.

This approach ensures that the GPU is never responsible for more than the memory cost of a single forward pass, plus a small overhead for the cached activations, if gradient check-pointing is not used to recompute activations. For the experiments reported in 7-8B scale models, all gradient-related computations were performed in 32-bit, though 16-bit gradients are equally possible. We employ the 16-bit strategy for our experiments with the Qwen2.5-32B-Instruct model in Appendix H.3.

We note that this memory overhead can be further reduced by dividing a model into parts and computing gradients separately but we have not implemented this optimization. We also note that we do not implement gradient check-pointing for the experiments presented in this paper. The above section describes the engineering setup used for the computation of gradients for the majority of the experiments presented in this paper, and decomposition optimizations were validated to produce the same outliers as with standard gradient collection procedures. Released code should reflect future changes to the engineering process if any significant changes are made.

### E.4 More details on quantization implementation.

TACQ uses GPTQ (Frantar et al., 2023a) for the quantization process of weights. In order to find the optimal scale for quantization in Eq. (1) we use a linear search to reduce the squared error for maximum change in weights introduced in QuaRot (Ashkboos et al., 2024). After the identification of salient weights, we use GPTQ to quantize the model while keeping the outliers in 16-bits. Specifically, prior to GPTQ, we set all salient weights to zero such that GPTQ ignores them during the quantization process, we then add back these weights post quantization. We employ GPTQ without considering outliers to simulate quantization for QAL. To find the memory cost of storing outlier weights, we follow SqueezeLLM and SPQR to estimate the cost as 32-bits per outlier when storing in the compressed sparse row format (Kim et al., 2024; Dettmers et al., 2023). Inference can be done using the SPQR kernel (Dettmers et al., 2023) or any other kernel that supports mixed-precision outliers. We do not employ double quantization, dynamic bit assignments, or activation reordering.

### E.5 Inducing Task Conditioning

We obtain gradient information from backpropagating the standard cross-entropy loss, which allows us to condition $\left|\frac{\partial \mathcal{L}}{\partial W_{ij}}\right|$ on both downstream tasks and standard calibration datasets such as C4 (Raffel et al., 2020). We condition the GPTQ component of TACQ on the same data used to generate the gradient.

In order to induce task conditioning, edits to GPTQ had to be implemented which required a nontrivial understanding of the codebase and underlying implementation. The GPTQ

conditioning mechanisms were designed for samples of the same length and padding samples when they are shorter significantly lowers the efficacy of the GPTQ compensation algorithm, therefore, edits must be made to allow examples of different lengths. SliM-LLM and SPQR use GPTQ as a backbone. To induce task conditioning in other methods we apply the same edits to the baselines as in our method. The necessity of this edit and the prevalence of methods requiring this edit (Ashkboos et al., 2024; Dettmers et al., 2023; Huang et al., 2024b) is evidence that the conditioning on downstream datasets was rarely considered. We include our edited version of GPTQ in our released code, which can be compared to the official GPTQ implementation for a full specification of the edits.

The official implementation of GPTQ used in TACQ and GPTQ based baselines sometimes fails in low bit scenarios. This is due to a failure in the process of computing a Cholesky decomposition to find the inverse Hessian in Eq. (10). We follow GPTQ (Frantar et al., 2023a) and add positive constants to dampen the diagonal of the Hessian and ensure the positive semi-definiteness of the Hessian prior to the decomposition, utilizing a progressive addition process that increases dampening until the decomposition is performed without errors. This is a standard fix which is an expansion of a static method for dampening described in Frantar et al. (2023a).

## F    Further Details on Evaluation and Conditioning Datasets

We note below the average number of calibration samples to fulfill the 128 sample x 2048 sequence length token budget. We define samples according to the definition with respect to the dataset. We note that for Spider, we use a zero-shot prompt to test performance in that setting, and thus use a larger number of examples. However, we use the same number of tokens as examples are shorter.

| Conditioning Dataset | Number of Examples |
| --- | --- |
| MMLU Full | 360 |
| MMLU STEM | 475 |
| MMLU Humanities | 249 |
| MMLU Social Sciences | 520 |
| Spider Text to SQL | 1520 |
| GSM8K Math | 318 |

Table 12: Number of examples per dataset to achieve equivalent token count to a standard 128 sample x 2048 sequence length C4 conditioning dataset.

## G    SPQR Pareto Comparison

For SPQR (Dettmers et al., 2023), six hyperparameters are tunable and the paper did not explicitly state hyperparameters for near 2-bit and 3-bit settings. Therefore, we adapt the GPTQ component of our method to use groupsize 16 to match one of SPQR's settings. We choose to adapt our method to groupsize 16 rather than adapt SPQR to our channel-wise setting as small groupsizes are stated to be an integral part of the SPQR strategy due to their observation that important weights may exist in small sensitive groups such as partial rows or attention heads. Specifically, we use group 16 and keep quantization scales in 16 bit. Then we adjust the percentage of outliers for each method to show that our method performs better across different outlier percentages and generalizes to small group quantization as used in SPQR. Fig. 4 shows the compression tradeoff and that our method is on the Pareto frontier. We note that this is not a practical setting for either TACQ or SPQR in terms of compression ratio and we create the setting in order to facilitate an apples to apples comparison in terms of hyperparameters.

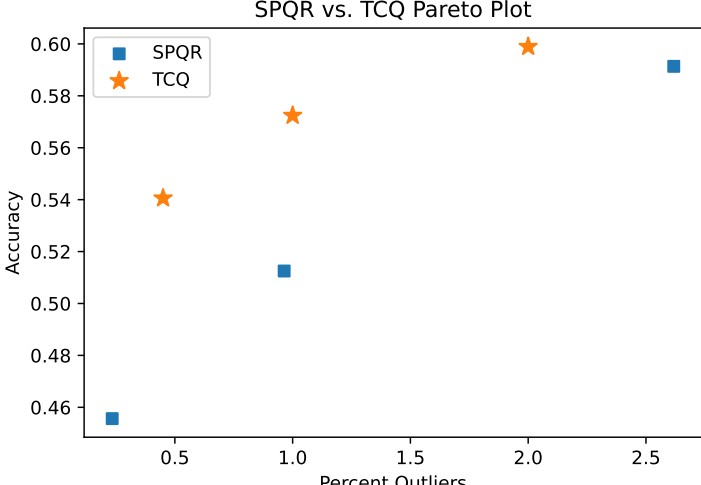

Figure 4: Comparison of accuracies on GSM8k at different compression ratios for Llama-3-8B-Instruct.

## H   Additional Results

### H.1   Qwen2.5-7B-Instruct Results

In Table 13, we replicate Table 2 for Qwen2.5-7B-Instruct, holding out .35% outliers for TACQ resulting in an average of around +.10 bit-width and using the same baseline methods as specified in Appendix A. We observe similar patterns to Llama-3-8B-Instruct with TACQ outperforming in all datasets at the 2-bit setting and for most datasets in the 3-bit setting.

### H.2   Llama-3-8B Perplexity Results

For perplexity, we evaluate on Llama-3-8B by conditioning on the C4 dataset as in Frantar et al. (2023a) with 128 samples of length 2048 and evaluating on the C4, WikiText, and PTB datasets. We observe that TACQ achieves better perplexity scores than all other methods.

### H.3   Model Size Generalization Test

We test the generalization of TACQ by evaluating it on the 4.6x larger Qwen2.5-32B-Instruct model, comparing it to the strongest baseline SliM-LLM in the 2-bit setting and presenting our results in Table 1. We observe that TACQ recovers 87.93% of the 16-bit unquantized model's performance at 2-bit quantization, reducing the model size by around 8x and outperforming SliM-LLM by 10.00%.

### H.4   Number of Datapoints for Gradient Capture

TACQ relies on a calibration dataset; here, we compare different sizes of this calibration dataset. Specifically, we test TACQ using varying numbers of calibration points for the computation of the gradient, presenting results in Table 15. The number of points is set by selecting datapoints from MMLU to fit the number of tokens for a fixed number of examples from C4, as in Section 3. The performance of the metric is relatively stable even when using few datapoints for the gradient capturing step. This aligns with similar observations by previous methods that have utilized gradient information in different ways (Kim et al., 2024; Shao et al., 2024b). We note that this feature can lead to time savings, with smaller calibration datasets requiring less time to condition on.

| | | | **2-bit** | | | |
|---|---|---|---|---|---|---|
| | | | **MMLU Splits** | | | |
| **Method** | **Avg bits** | **GSM8k** | **Social Sciences** | **STEM** | **Humanities** | **Full** |
| Full | 16 | 80.97 | 83.36 | 68.87 | 69.01 | 73.90 |
| GPTQ | 2.00 | 4.55 | 44.13 | 38.39 | 40.39 | 38.17 |
| SliM | 2.125 | 29.80 | 54.19 | 43.67 | 43.27 | 47.89 |
| Squeeze | 2.15 | 6.52 | 37.94 | 39.31 | 35.48 | 36.55 |
| SPQR | 2.15 | 2.05 | 24.52 | 26.65 | 25.74 | 25.33 |
| TACQ | 2.10 | **47.08** | **66.58** | **49.47** | **51.23** | **56.13** |

| | | | **3-bit** | | | |
|---|---|---|---|---|---|---|
| | | | **MMLU Splits** | | | |
| **Method** | **Avg bits** | **GSM8k** | **Social Sciences** | **STEM** | **Humanities** | **Full** |
| Full | 16 | 80.97 | 83.36 | 68.87 | 69.01 | 73.90 |
| GPTQ | 3.00 | 67.55 | 79.48 | **66.62** | 63.42 | 69.14 |
| SliM | 3.125 | 76.19 | 79.10 | 63.06 | 67.06 | 70.42 |
| Squeeze | 3.145 | 75.97 | 78.71 | 63.59 | 62.23 | 69.25 |
| SPQR | 3.14 | 62.85 | 80.26 | 63.32 | 65.45 | 67.87 |
| TACQ | 3.12 | **78.32** | **80.65** | 66.49 | **67.82** | **71.64** |

Table 13: Comparison of downstream task accuracy (%) across quantization methods on Qwen2.5-7B-Instruct.

| **Method** | **Bits** | **C4** | **WikiText2** | **PTB** |
|---|---|---|---|---|
| **Base (no quant)** | 16 | 9.44 | 6.14 | 11.18 |
| **GPTQ** | 2.00 | 136.62 | 410.63 | 288.26 |
| | 3.00 | 16.41 | 12.28 | 18.68 |
| **SPQR** | 2.15 | 4.14e5 | 5.95e5 | 4.73e5 |
| | 3.15 | 14.38 | 9.28 | 15.24 |
| **SliM** | 2.125 | 96.58 | 248.06 | 245.27 |
| | 3.125 | 12.53 | 8.15 | 13.29 |
| **Squeeze** | 2.15 | 287.54 | 195.39 | 271.58 |
| | 3.15 | 12.85 | 7.93 | 13.68 |
| **TACQ** | 2.10 | 25.19 | 17.66 | 28.20 |
| | 3.10 | 11.81 | 7.53 | 12.57 |

Table 14: Perplexities on C4, WikiText2, and PTB for Llama-3-8B and its quantized variants. Lower perplexity is better.

| Equivalent C4 Examples | MMLU Datapoints | MMLU Accuracy (%) |
|:---:|:---:|:---:|
| 16 | 48 | 63.16 |
| 64 | 179 | 63.70 |
| 128 | 360 | 63.28 |

Table 15: MMLU accuracy when using different number of datapoints for gradient capture. Results based on quantizing Llama-3-8B-Instruct with TACQ at 3-bits with bit-width 3.10 using .35% outliers. We report both the number of MMLU datapoints used as well as how many C4 calibration datapoints this is equivalent to in terms of the number of tokens used. Following GPTQ (Frantar et al., 2023a), each C4 example is defined as a sequence of 2048 tokens.

