# OpenReview forum: "Task-Circuit Quantization: Leveraging Knowledge Localization and Interpretability for Compression"
_colmweb.org/COLM/2025/Conference — COLM 2025_

### Official Review · Reviewer_DkhF · 2025-05-12

**Rating:** 9
**Confidence:** 5
**Ethics Flag:** 1

**Summary:**

The paper presents Task-Circuit Quantization (TCQ), a mixed-precision PTQ approach that retains approximately 0.35 % of weights in 16-bit format while quantizing the remainder to 2- or 3-bit precision. TCQ differs from prior outlier-based schemes by defining a task-conditioned saliency measure: a quantization-aware localization term estimates the deviation each weight would experience under uniform GPTQ quantization, and a magnitude-weighted gradient term reflects the weight’s importance to the downstream objective. The product of these two scores is computed with a single backward pass over a 128-sample calibration set to decide which weights remain in higher precision. Empirical evaluations on Llama-3-8B-Instruct and Qwen-2.5-7B-Instruct indicate that TCQ attains 97 % of the original 16-bit accuracy on MMLU with an effective precision of 3.1 bits, and it achieves higher accuracy than Slim-LLM, SPQR, and SqueezeLLM at 2-bit precision on MMLU, GSM8k, and Spider. Ablation studies show that removing either component of the saliency score or omitting task-specific calibration data degrades performance, particularly at ultra-low bit widths.

**Questions To Authors:**

Compute footprint: What wall-clock time and peak GPU memory does TCQ require for an 8 B model—and how would this scale to a larger model? A concrete comparison with GPTQ and Slim-LLM would help.

Inference efficiency: Could you provide latency or throughput measurements (e.g., tokens/sec on a GPU) for the mixed-precision kernels used, so readers can assess real-world gains beyond accuracy?

Outlier ratio sensitivity: Performance is reported for a fixed 0.35 % 16-bit budget; how does accuracy trade off against memory when the preserved-weight fraction is swept from 0.1 % to 2 %?

**Reasons To Accept:**

The paper offers a conceptually fresh bridge between mechanistic-interpretability work on circuit localization and the practical goal of weight compression. By explicitly modelling how each weight’s value will shift during quantization and multiplying this with a task-conditioned gradient signal, TCQ provides an alternative to magnitude- or Fisher-based heuristics. The method is simple to implement (one hyper-parameter: outlier ratio), slots cleanly on top of existing GPTQ code, and—most importantly—produces state-of-the-art accuracy in both 2- and 3-bit settings while using fewer high-precision weights than competing schemes. The experimental section is broad (two model families, three diverse tasks, calibration-set transfer, saliency ablations) and includes qualitative generation examples that highlight the practical benefits of preserving instruction-following ability.

**Reasons To Reject:**

Despite its strong empirical showing, my main criticism of the paper is it's lack of scale results. All evaluations stop at the 8 B-parameter scale; it is unclear whether capturing gradients for 34 B/70 B models remains tractable or whether the saliency signal remains sparse enough. The study also focuses almost exclusively on accuracy; latency, throughput and kernel efficiency of the resulting mixed-precision models—necessary for on-device deployment—are not reported.

---

> ### Author Response · Authors · 2025-05-30
> **Response to Reviewer DkhF**
>
> Thank you for your review and for appreciating our “broad” experimental section, “one hyper-parameter” setup, and building a “conceptually fresh bridge” between mechanistic-interpretability and quantization. We proceed to address the remaining comments in your review:
>
> **Scaling to Larger Models.**
> Thanks for this suggestion -- we have added an experiment to show that TCQ continues to outperform GPTQ [2] and SliM-LLM [3] in the 4.6x larger Qwen2.5-32B-Instruct model. We observe that TCQ recovers 87.93% of the unquantized model’s performance at 2-bit quantization, reducing model size by 8x and outperforming Slim-LLM by 10.00% and GPTQ by 21.96%. Additionally, TCQ quantization remains tractable on a single 4x NVIDIA RTX A6000 48G node in 6.93 hours. Here we use 16 bit gradients for TCQ.
>
> Performance on Qwen2.5-32B-Instruct
> | Method       | Avg bits | MMLU Accuracy % |
> |-|-:|-:|
> | **Unquantized** | 16.00  | 82.46 |
> | **GPTQ**    | 2.00  | 50.55 |
> | **SliM-LLM**    | 2.125  | 62.51 |
> | **TCQ**        | 2.10   | 72.51 |
>
> **Q1. Compute Footprint.**
> We briefly covered quantization compute footprint in Appendix A. Based on your suggestion we expand the comparison to GPTQ, SliM-LLM and TCQ for 7B and 32B in the table below and will include it in our final paper. We compute these numbers for all methods in the setting of 2-bit quantization conditioned on MMLU_MCQA. *TCQ quantization is significantly faster than SliMLLM for both 7B and 32B models.*
>
> | Method | Compute Time (hours) | GPU Memory Peak (GB) |
> |-|-|-|
> |GPTQ 7B| .58 | 7.68 |
> |SliMLLM 7B| 14.66 | 12.48 |
> |TCQ 7B| 2.76 + .58 | 93.29 |
>
> | Method | Compute Time (hours) | GPU Memory Peak (GB) |
> |-|-|-|
> |GPTQ 32B| 2.56 | 20.16 |
> |SliMLLM 32B| 54.71 | 22.608 |
> |TCQ 32B| 4.37 + 2.56| 138.62|
>
> While gradient capture is expensive, there are several reasons why we believe it is tractable in the context of TCQ. We also conduct new experiments to show methods we can use to drastically reduce the cost of gradient capture by 8x without affecting performance.
> - The number of datapoints used for gradient capture can be reduced by 8x without affecting performance. See Table A below.
> - Gradients only need to be captured once to freely quantize at varied % outliers and base bitness.
> - In Appendix D, line 708-733, we detail a memory saving trick for gradient capture that offloads the last stage of gradient computation to the CPU, and suggest other possible engineering improvements.
> - Previous works such as SqueezeLLM [5] and GWQ [4], which we compare against both require gradient capture and assume reasonable cost.
>
> Table A: varying the number of calibration examples for gradient capture when compressing Llama-3-8B-Instruct to 3 bits on MMLU.
> | Equivalent Calibration Examples | MMLU Accuracy (%) |
> |-:|-:|
> | 16                    | 63.16 |
> | 64                    | 63.70 |
> | 128                  | 63.28 |
>
> **Q2. Inference Efficiency.**
> In terms of inference efficiency we note that our framework produces weights in the format of SPQR’s setup [1], and inference times should be the same by using SPQR’s kernel. We note this in Appendix D.2 line 758. We view our main contributions as the TCQ metric and task-specific quantization method, and thus do not evaluate using a kernel. Details on inference speeds can be found in Table 4 of the SPQR paper [1].
>
> **Q3. Outlier Ratio Sensitivity.**
> In Figure 3, we show a pareto plot where we evaluate TCQ with higher outlier percentages, showing the accuracy memory tradeoff as you increase the number of outliers used. We note that 2% outliers correspond to approximately .75 additional bit-width. In general, a higher outlier ratio is not advisable in terms of memory use. Noted in line 542, average bit-width can be calculated as bit-width = N − Nr + 32r where r is the ratio of outliers we use. Appendix F, Figure 4, contains an additional pareto comparison where we sweep outlier ratio from .5% to 2%.
>
> [1] Dettmers et. al., SpQR: A SparseQuantized Representation for Near-Lossless LLM Weight Compression, June 2023. URL
> http://arxiv.org/abs/2306.03078. arXiv:2306.03078.
>
> [2] Frantar et. al., GPTQ: Accurate PostTraining Quantization for Generative Pre-trained Transformers, March 2023a. URL
> http://arxiv.org/abs/2210.17323. arXiv:2210.17323 [cs].
>
> [3] Huang et. al., SliM-LLM: Salience-Driven Mixed-Precision Quantization
> for Large Language Models, May 2024b. URL http://arxiv.org/abs/2405.14917.
> arXiv:2405.14917 [cs].
>
> [4] Shao et. al., GWQ: Gradient-Aware Weight Quantization for
> Large Language Models, December 2024b. URL http://arxiv.org/abs/2411.00850.
> arXiv:2411.00850 [cs].
>
> [5] Kim et. al., SqueezeLLM: Dense-and-Sparse Quantization,
> June 2024. URL http://arxiv.org/abs/2306.07629. arXiv:2306.07629 [cs].

---

> > ### Comment · Reviewer_DkhF · 2025-06-01
> > **Thanks for the new set of experiments!**
> >
> > The reviewer would like to thank the authors for conducting the experiments as requested. The reviewer would like to reiterate the fact that they think this is a strong paper and have increased their score thus.

---

> > > ### Author Response · Authors · 2025-06-04
> > > **Followup to Reviewer DkhF**
> > >
> > > Thank you for suggesting the scaling and compute footprint experiments, which has improved the paper. We’re glad that you thought the new experiments were satisfactory, and thank you for raising your score.

---

### Official Review · Reviewer_zLzX · 2025-05-14

**Rating:** 5
**Confidence:** 4
**Ethics Flag:** 1

**Summary:**

In this paper, the authors propose Task-Circuit Quantization, a mixed-precision approach to reduce model memory consumption without dropping much downstream model performance. Specifically, they leverage task-specific calibration data for knowledge localization and achieve interpretability. Experiments on several standard datasets show that the proposed method can achieve better downstream task accuracy on Llama-3-8B-Instruct. They also did several analysis like conditioning transfer and ablation to demonstrate the effectiveness of the proposed method.

**Reasons To Accept:**

* Interesting and reasonable method to conduct quantization to compress LLMs.
* Both QAL and MSG play important roles in the framework per the ablation study.
* Better downstream performance compared to baseline methods with a similar compression rate.

**Reasons To Reject:**

* Some recent works like GWQ (Shao et al., 2019) and QAT (Ma et al 2024) are not compared in the experiments.
* Lack of concrete examples to show how QAL and MSG work.
* It would also be good to evaluate the performance on language modeling, even it is designed for certain tasks.

---

> ### Author Response · Authors · 2025-05-30
> **Response to Reviewer zLzX**
>
> Thank you for your review and for pointing out that TCQ is a “Interesting and reasonable method to conduct quantization”. We proceed to address your comments below.
>
> **Q1a: GWQ comparison.**
> In section 4.5 and Table 5 of the paper, we do in fact mention and compare against the GWQ baseline [4]. We labeled this as “Sample Absolute Gradient” in Table 5 due to the fact that GWQ’s metric is equivalent to the gradient term of the TCQ metric. Thus, we already compare our TCQ outlier selection metric in an apples-to-apples setting to GWQ’s outlier selection metric. The TCQ metric outperforms GWQ metric (Sample Absolute Gradient) by 6.49% at 2 bits and 1.81% at 3 bits, evaluated on MMLU accuracy. Thanks for the comment, we will clarify this in the final version.
>
> **Q1b: QAT comparison.**
> We propose a Post-Training-Quantization method and following SqueezeLLM [1], GPTQ [2], and Slim-LLM [3], view a comparison to QAT as out of scope. Quantization-Aware-Training’s increased cost -- requiring significantly more data points and fully training the model for many steps -- means that past work has treated it as a separate category.
>
> **Q2: QAL and MSG concrete examples.**
> Concretely, we expect that when QAL is used, the average distance to gridlines will increase. Intuitively, given the same gradient importance, it is more valuable to preserve outliers far away from gridlines as they will be more heavily impacted when quantized. We compute this average distance metric and show that this is indeed the case.
>
> | Scale | Gradient | QAL |
> |----|----|----|
> | Outlier distance to gridlines | 3.426e-3 ± 1.604e-3 | 9.577e-3 ± 4.643e-3 |
>
> This new experiment was conducted on Llama-3-8B-Instruct, comparing the average distance to gridlines for outliers identified with just the gradient to those identified with QAL. In the final version of the paper, we will plan to include this result in an expanded discussion on the concrete impacts of QAL and MSG.
>
> **Q3: Perplexity results.**
> Thanks for pointing this out – we in fact include these results in Appendix G, Table 9, where we show perplexity results indicating that TCQ outperforms other methods in terms of language modeling across wikitext2, c4_new, and ptb_new datasets. We also point to this table in section 4, Datasets for Quantization Conditioning and Evaluation, line 207. We’ll move this pointer to the start of section 4 in the final version.
>
> We include an updated version of these results on Llama-3-8B quantized with C4 as the conditioning dataset here for quick reference:
> | Method | Bits | C4 | WikiText-2 | PTB |
> |--------|------|------|-----------|------|
> | **Base&nbsp;(no&nbsp;quant)** | 16   | 9.44   | 6.14   | 11.18 |
> | **GPTQ** | 2.00 | 136.62 | 410.63 | 288.26 |
> |          | 3.00 | 16.41  | 12.28  | 18.68 |
> | **SPQR** | 2.15  | 4.14e5 | 5.95e5 | 4.73e5 |
> |          | 3.15  | 14.38  | 9.28   | 15.24 |
> | **Slim** | 2.125 | 96.58  | 248.06 | 245.27 |
> |          | 3.125 | 12.53  | 8.15   | 13.29 |
> | **Squeeze** | 2.15 | 287.54 | 195.39 | 271.58 |
> |            | 3.15 | 12.85 | 7.93  | 13.68 |
> | **TCQ** | 2.10 | 25.19 | 17.66 | 28.20 |
> |          | 3.10 | 11.81 | 7.53  | 12.57 |
>
> Note that this table differs slightly from Table 9, in that it includes Slim-LLM (which we initially excluded due to its long runtime).
>
> [1] Sehoon Kim, Coleman Hooper, Amir Gholami, Zhen Dong, Xiuyu Li, Sheng Shen,
> Michael W. Mahoney, and Kurt Keutzer. SqueezeLLM: Dense-and-Sparse Quantization,
> June 2024. URL http://arxiv.org/abs/2306.07629. arXiv:2306.07629 [cs].
>
> [2] Elias Frantar, Saleh Ashkboos, Torsten Hoefler, and Dan Alistarh. GPTQ: Accurate PostTraining Quantization for Generative Pre-trained Transformers, March 2023a. URL
> http://arxiv.org/abs/2210.17323. arXiv:2210.17323 [cs].
>
> [3] Wei Huang, Haotong Qin, Yangdong Liu, Yawei Li, Xianglong Liu, Luca Benini, Michele
> Magno, and Xiaojuan Qi. SliM-LLM: Salience-Driven Mixed-Precision Quantization
> for Large Language Models, May 2024b. URL http://arxiv.org/abs/2405.14917.
> arXiv:2405.14917 [cs].
>
> [4] Yihua Shao, Siyu Liang, Zijian Ling, Minxi Yan, Haiyang Liu, Siyu Chen, Ziyang Yan,
> Chenyu Zhang, Haotong Qin, Michele Magno, Yang Yang, Zhen Lei, Yan Wang, Jingcai Guo, Ling Shao, and Hao Tang. GWQ: Gradient-Aware Weight Quantization for
> Large Language Models, December 2024b. URL http://arxiv.org/abs/2411.00850.
> arXiv:2411.00850 [cs].

---

> > ### Author Response · Authors · 2025-06-04
> > **Friendly Reminder for Reviewer zLzX**
> >
> > As the second discussion section is now underway, we wanted to briefly ping to ask whether our response has addressed your comments and allowed you to revisit your score. We want to especially highlight our response above on GWQ and on language modeling, for which we point to the sections/tables in the paper these results can be found, as well as our new analysis on QAL and gridline distance in the rebuttal.

---

> > ### Author Response · Authors · 2025-06-09
> > **Nearing End of Discussion Period Reminder for Reviewer zLzX**
> >
> > Dear Reviewer zLzX,
> >
> > Given that there is only 1 day remaining in the discussion period (ending on June 10th), we wanted to briefly ping again to see if our response has addressed your comments. We are happy to address any questions you might have in the time that remains.

---

### Official Review · Reviewer_FNLA · 2025-05-21

**Rating:** 7
**Confidence:** 3
**Ethics Flag:** 1

**Summary:**

This paper presents Task-Circuit Quantization (TCQ), a novel mixed-precision post-training quantization method for large language models (LLMs). TCQ identifies task-specific "circuits"—critical weight subsets vital for downstream tasks—by combining Quantization-Aware Localization (QAL) and Magnitude-Sharpened Gradient (MSG) . While QAL is used to estimate quantization-induced weight changes' impact on loss, MSG is for assessing weight importance via gradient and magnitude. By preserving the top ~0.35% most salient weights in 16-bit precision while quantizing others to 2-3 bits, TCQ maintains performance with minimal memory overhead. Experiments on Llama-3 and Qwen2.5 models across QA, math reasoning, and text-to-SQL tasks show TCQ outperforms baselines like SPQR and Slim-LLM, achieving up to 14.74% absolute accuracy gains in 2-bit settings and recovering 97% of unquantized MMLU performance at 3.1 bits. TCQ’s simplicity (one hyperparameter) and effectiveness in ultra-low bit-width compression make it suitable for edge deployments and task-specific LLM optimization.

**Reasons To Accept:**

This paper propose a novel idea for mixed-presicion post-training quantization by leveraging concepts from mechanistic interpretability (e.g., circuit discovery and gradient attribution).  By combining QAL and MSG, this dual approach ensures that TCQ preserves weights that are both sensitive to quantization and critical for task performance, rather than relying on heuristic or generic saliency metrics.

Experiments are conducted on multiple tasks against several baselines, and the results show it significantly outperforms the baselines.

The paper is well-structured, with a clear problem statement, motivated methodology, and comprehensive evaluation and ablation study.

**Reasons To Reject:**

The paper is good in general.  I do not see specific reasons to reject it.

---

> ### Author Response · Authors · 2025-05-30
> **Response to Reviewer FNLA**
>
> We thank you for your review, and are glad that you thought our paper was “well-structured” and had a “motivated methodology” and “comprehensive evaluation and ablation” studies.

---

### Official Review · Reviewer_WCni · 2025-05-26

**Rating:** 7
**Confidence:** 4
**Ethics Flag:** 1

**Summary:**

This paper presents a method for quantizing LLMs on a per-task basis. The method uses a small calibration set per dataset and a novel "TCQ" method for determining which weights to preserve and which to quantize.  As presented, this method combines existing ideas from attribution patching as well as saliency measures ("magnitude-sharpened gradients") from the interpretability literature. In mixed-precision settings down to 2-bit and 3-bit, the method shows decent performance across several benchmark datasets, outperforming existing approaches.

**Questions To Authors:**

Table 1 should make it clearer which systems share quantization and which don't. As I understand it, TCQ is different for every column but baselines are the same? Table 3 seems to mitigate this by showing a clear experiment of conditioning different methods on different sets, but if some of the benefit is derived from additional, specific conditioning of TCQ, that should be made clear.

I also have a question about the following:

> importance for weights close to values that can be represented after quantization (i.e. values on the gridlines) is reduced to near zero, even if the gradients of these weights are large and the weights are generally important to the network.

Scaling by |W_ij| (the additional term in TCQ) doesn't obviously fix this issue for moderate weights. I'm also not sure why this is an issue. If the approximation from quantization is close to exact, why is it important for the weight to be assigned a high importance? Figure 2 would seem to indicate that such values close to the gridline can be quantized.

**Reasons To Accept:**

The paper tackles an interesting and timely problem. I think quantizing LLMs for specific applications is of broad enough interest, even if it adds extra overhead compared to "general" quantization methods.

The method is simple and intuitive. The quantization formula does not introduce extra moving pieces beyond the baselines except for the need for additional data.

The paper compares against several recent baselines and shows gains across a number of tasks.  In my view, the results are fairly strong. Table 3 indicates that the specialization to a dataset is not too narrow, and the method outperforms others even when evaluated on other datasets.

The paper is well-written.

**Reasons To Reject:**

The use of "mechanistic interpretability" in this paper, and the connections to circuits, is questionable. Equation (5) can be arrived at without any motivation from activation patching or comparing clean and corrupted models; it's just a first-order Taylor approximation of the loss incurred by quantization. The connection to circuits is unclear: there is no explicit discovery of circuits here and no second-order interaction modeling which I would expect from an approach that considers circuits in a deep way. As a result, the motivation of the method is a bit lacking, and I don't have a strong sense that this is necessarily the best method of this form (e.g., further modifications of the formula in equation (7) could work better).

---

> ### Author Response · Authors · 2025-05-30
> **Response to Reviewer WCni**
>
> Thank you for your careful review of our paper, and for highlighting our “simple and intuitive” approach to an “interesting and timely problem”.
>
> **Mechanistic Interpretability Connection.**
> To clarify our connection to Mechanistic Interpretability, we trace the line of MI ideas motivating our work. Activation Patching, originally introduced in ROME [1] for localization and model editing, was applied for localization of relevant model components. Attribution Patching [2] was introduced as a gradient based reformulation of Activation Patching with significant efficiency improvements, and applied to automated circuit discovery [3, 4]. We take the efficiency insight and formulation from Attribution Patching and apply it for localization of task relevant weights, and define what we find as “weight circuits” to separate them from activation centric definitions of circuits. Nevertheless, we recognize that circuits can refer to manually discovered circuits and more intricate edge based processes and will include an extended discussion of this in our final paper. To that end, we would appreciate any elaboration on examples of “second-order interaction” that might be helpful to consider.
>
> **Question 1: which systems “share quantization”.**
> All methods are conditioned in Table 1. Specifically, all methods we compare against depend on calibration data for quantization, and have their conditioning data replaced for each evaluation dataset. Additionally, with the exception of the uniform precision GPTQ, all methods use calibration data to determine their mixed precision outlier selection or layout, which we also replace. For example, for Slim-LLM, each seed in each square of Table 1 consisted of a ~12-18 hour run where calibration was conditioned on the specific dataset and specific seed. In Table 3, no additional conditioning of TCQ was done beyond that for other methods, in order to show the generality of our method. We thank you for pointing this out and will emphasize our apples-to-apples conditioning setup in the final version of this paper.
>
> **Question 2a: MSG for moderate weights.**
> We thank you for the question. By “moderate weights”, we assume you are referring to weights where |W_ij| is small (please let us know if meant something else by this), in which case upweighting by |W_ij| would not fix approximation errors from QAL. However, as 0 is always a gridline and our quantization implementation uses a simple linear search strategy to shrink the quantization scales (discussed in lines 748-751), the majority of weights with small |W_ij| are already fairly well captured by gridlines, especially compared with weights with large |W_ij|. We briefly discuss this in Appendix D, Expanded Description of TCQ, lines 691-702.
>
> **Question 2b: exactness of QAL approximation.**
> While we affirm that using a small percent of outliers makes the QAL estimate “informative”, it is not perfect. We note that we do not preserve the same ratio of outliers for each weight matrix. For example, when quantizing Qwen2.5-7B-Instruct, the 27th mlp down projection preserves 5.226e+04 outliers, while the 18th mlp down projection layer preserves an order of magnitude more outliers at 3.604e+05. The distribution is also skewed in a channel wise manner. We observe that in channels where the number of outliers is large, the gridlines can be altered. In those cases, MSG acts as a stabilizing agent by prioritizing a gridline-agnostic importance.
>
>
> [1] Kevin Meng, David Bau, Alex Andonian, and Yonatan Belinkov. Locating and Editing
> Factual Associations in GPT, January 2023. URL http://arxiv.org/abs/2202.05262.
> arXiv:2202.05262 [cs].
>
> [2] Neel Nanda. Attribution Patching: Activation Patching At Industrial Scale, 2022. URL
> https://www.neelnanda.io/mechanistic-interpretability/attribution-patching.
>
> [3] Janos Kramar, Tom Lieberum, Rohin Shah, and Neel Nanda. AtP*: An efficient and
> scalable method for localizing LLM behaviour to components, March 2024. URL http://arxiv.org/abs/2403.00745. arXiv:2403.00745 [cs].
>
> [4] Aaquib Syed, Can Rager, and Arthur Conmy. Attribution Patching Outperforms Automated Circuit Discovery, November 2023. URL http://arxiv.org/abs/2310.10348.
> arXiv:2310.10348 [cs].

---

> > ### Comment · Reviewer_WCni · 2025-06-04
> > **Thanks for the clarification**
> >
> > Second-order effects: I was thinking of approaches like path patching (e.g., used here https://arxiv.org/pdf/2402.14811 ). However, I take your point that your notion of circuits is in weight space rather than activation space. I still feel there is a clearer formulation of what you do that doesn't draw on mech interp terminology at all but instead uses a more optimization-centric framing (the Taylor approximation motivation that I stated). However, I'll offer this as soft guidance as it's not a major sticking point in my assessment.
> >
> > My overall score is unchanged.

---

> > > ### Author Response · Authors · 2025-06-04
> > > **Followup Response to Reviewer WCni**
> > >
> > > Thank you for your clarifying pointer to path patching methods. Yes, we do not explore these types of edge based interactions but focus on other interpretations. Specifically, we did not focus on path-patching methods due to their $O(n^2)$ complexity, required for connecting every node to every subsequent node through the residual stream. This is infeasible in our case due to the need to capture patterns at the parameter level. In the final paper, we will plan to define the notion of circuits and weight circuits earlier during the Methods section, and attempt to incorporate your other feedback to improve our discussion of optimization and mechanistic interpretability concepts.

---

### Decision · Program_Chairs · 2025-07-08

**Decision:**

Accept

**Comment:**

The paper proposed a novel and simple PTQ technique that is a combination of techniques across various streams of research that showed promising results. Most the reviewers agree that it was a well written paper with clear technical description and empirical wins over the baselines. While the fundamentals remain same across techniques, the elegant combination of ideas makes this paper stand out.

After going through the paper, reviews and rebuttal, I recommend accepting the paper and it adds value to the community.

[Automatically added comment] At least one review was discounted during the decision process due to quality]